# Alzheimer’s Disease: Challenges and a Therapeutic Opportunity to Treat It with a Neurotrophic Compound

**DOI:** 10.3390/biom12101409

**Published:** 2022-10-02

**Authors:** Narjes Baazaoui, Khalid Iqbal

**Affiliations:** 1Biology Department, College of Sciences and Arts Muhayil Assir, King Khalid University, Abha 61421, Saudi Arabia; 2Department of Neurochemistry, Inge Grundke-Iqbal Research Floor, New York State Institute for Basic Research in Developmental Disabilities, 1050 Forest Hill Road, Staten Island, NY 10314, USA

**Keywords:** Alzheimer’s disease, amyloid beta, neurodegeneration, neurogenesis, neuroinflammation, neuronal plasticity, neurotrophic factors, peptide-based therapeutics, P021, synaptic deficit, tau pathology

## Abstract

Alzheimer’s disease (AD) is a progressive neurodegenerative disease with an insidious onset and multifactorial nature. A deficit in neurogenesis and synaptic plasticity are considered the early pathological features associated with neurofibrillary tau and amyloid β pathologies and neuroinflammation. The imbalance of neurotrophic factors with an increase in FGF-2 level and a decrease in brain derived neurotrophic factor (BDNF) and neurotrophin 4 (NT-4) in the hippocampus, frontal cortex and parietal cortex and disruption of the brain micro-environment are other characteristics of AD. Neurotrophic factors are crucial in neuronal differentiation, maturation, and survival. Several attempts to use neurotrophic factors to treat AD were made, but these trials were halted due to their blood-brain barrier (BBB) impermeability, short-half-life, and severe side effects. In the present review we mainly focus on the major etiopathology features of AD and the use of a small neurotrophic and neurogenic peptide mimetic compound; P021 that was discovered in our laboratory and was found to overcome the difficulties faced in the administration of the whole neurotrophic factor proteins. We describe pre-clinical studies on P021 and its potential as a therapeutic drug for AD and related neurodegenerative disorders. Our study is limited because it focuses only on P021 and the relevant literature; a more thorough investigation is required to review studies on various therapeutic approaches and potential drugs that are emerging in the AD field.

## 1. Introduction

AD is the most prevalent neurodegenerative disease of aging which has a complex and multifactorial nature for which the different mechanisms involved remain to be fully understood. It affects one in eight older Americans, and it is characterized by progressive memory loss and cognitive impairment that severely affect the patient’s daily life [1,2]. Impairments can include difficulties in learning, deficient spatial recognition and slow cognitive performance [3,4]. The two major pathological hallmarks of the disease are the extracellular deposits of amyloid β(Aβ) as plaques [5] and the intracellular accumulation of hyperphosphorylated tau as neurofibrillary tangles [6,7,8]. Both Aβ and tau pathologies are preceded by synaptic and neuronal loss early in the disease development. Synaptic and neuronal losses take place in specific areas of the brain. Histological and imaging studies show that the entorhinal cortex (EC) is affected first, followed by spread to the hippocampus and the cerebral cortex [9,10,11]. Although the density of neurofibrillary tangles and not amyloid plaques in the brain correlates with dementia [12,13], it is very likely that additional features are implicated in the cognitive decline seen in AD patients. Neuroinflammation, for instance, is one of the other features which is defined as a specialized immune response that develops in the central nervous system and affects the process of neurogenesis in the hippocampus [9]. An imbalance in the neurotrophic factors and lack of intrinsic support is also proposed as another pathological feature of AD [14,15].

AD encompasses early and late onset types. The early onset familial form is rare, affects less than 5% of the cases, and is caused by mutations in either presenilin 1, 2 (PSEN1, PSEN2) or the amyloid precursor protein (APP) genes. Presenilin 1 (PS1) and presenilin 2 (PS2) proteins are found in the catalytic core of the γ-secretase complex and mutation in the PSEN gene facilitates amyloidosis [16,17]. In the late onset, APOε4 allele is considered as major risk factor [18,19]. However, a major burden that remains for drug development is that there is an ambiguity in the mechanisms underlying AD pathophysiology. The central dogma that Aβ plays a crucial role in the development of AD has led to many Aβ-based therapeutic clinical trials but to date they have been disappointing. Nearly all drug treatments available today cannot stop or even slow down the progression of the disease [20]. Since there is a great need to develop a drug that can slow and/or stop the progression of the disease, attention has been shifted to developing drugs that target multiple disease aspects. Furthermore, efforts are being made now to prevent AD at its early stages rather than at later stages of its progression [14]. Prevention could be primary or secondary. In our laboratory we focused on the development of a neurotrophic peptidergic drug that can mimic neurotrophins and promote neurogenesis, the maturation of newborn cells into mature functional neurons, and that can increase neuronal plasticity to prevent cognitive and memory decline. We have developed a peptidergic compound named P021 which is composed of four amino acids derived from a biologically active region of ciliary neurotrophic factor (CNTF) and is C-terminally adamantylated to increase its lipophililicity and decrease its degradation by exopeptidases. In this article we review recent studies that were conducted in our laboratory to evaluate its potential as a drug for the treatment of AD and other neurodegenerative diseases.

The major objectives of the present work are: (i) to introduce and summarize the major etiopathology features of AD, both the neuropathological as well as the psychological, (ii) provide an update about the current status of the development of a potential treatment of the disease, and (iii) to introduce a neurotrophic and neurogenic peptide mimetic compound that was developed in our laboratory and was found to overcome the difficulties faced in the administration of the whole neurotrophic factor proteins. The approaches that we used in the present work were to gather data from the literature and the major discoveries concerning the neuropathological and psychological features of AD and to describe the pre-clinical studies that were conducted in our laboratory concerning P021 and its potential use as a therapeutic drug for AD and related neurodegenerative disorders.

## 2. Ethiopathogenesis of AD

### 2.1. Amyloid β and Neurofibrillary Tangles

AD is a proteopathy, manifested through the formation of Aβ aggregates and neurofibrillary tangles (NFT) (Table 1) [21]. Aβ deposits are formed following the processing of amyloid precursor protein (APP) through the amyloidogenic pathway. The cleavage of APP by enzymes β-secretase (BACE-1) followed by γ-secretase results in the formation of Aβ. Under pathological conditions and in the case of AD, excessive formation of toxic Aβ oligomers and senile plaques does happen in the basal forebrain as well as in the cortex, hippocampus and amygdala [20,22,23,24]. Although Aβ plaques are considered the pathological hallmarks of the disease; the intracellular accumulation of small Aβ oligomeric species in the endoplasmic reticulum as well as extracellularly is thought to be the cause of toxicity in AD [24,25,26]. The protein aggregates damaging the cells with the smallest oligomers are considered the most neurotoxic either for Aβ or tau [21,27].

Intracellular NFT are composed mainly of hyperphosphorylated tau that start first in the entorhinal cortex (EC) and the hippocampus [9,11,14,28]. The hyperphosphorylated tau proteins that misfold adopt a high β-sheet content, leading to the formation of intraneuronal NFTs [21,36,37,38,39]. Indeed, protein monomers of tau become stacked in register and parallel to each other to form smaller soluble oligomers and consequently larger insoluble protofilaments that associate in an anti-parallel conformation to form protofibrils and eventually paired helical filaments (PHFs) or straight filaments found in NFTs and other tau inclusions [21,40,41].The formation of tau aggregates induces the intracellular accumulation of tau proteins in aggresomes [42], and the release of the oligomers of hyperphosphorylated tau from the neuron into the extracellular space is believed to lead to the spread and propagation of tau pathology across different brain regions [43].The density of NFT correlates the most with cognitive decline and neurodegeneration compared to Aβ deposition in AD [21,44]. Although the severity of dementia is strongly correlated with the density of NFT or NFT plus senile plaques, neither of the two pathologies is specific to AD. In the oldest old, for example, dementia is very weakly associated with both pathologies; the presence of high levels of Aβ and tau pathologies were reported to exist in the absence of dementia [45,46,47]. Furthermore, a significant overlap was reported in the BRAAK stages between demented and non-demented patients [47,48,49]. Hence, there must be other pathological features that are involved in the pathogenesis of AD, and that is what explains its complex and multifactorial etiology.

### 2.2. Relation between Key Player Proteins in AD Pathology and Plasticity

Both genes that are known as key players in AD, such as PSEN 1 and 2 and tau, are also physiologically involved in the plasticity of the brain at the embryonic stages, especially as membrane proteins and when concentrated in synapses.

#### 2.2.1. Presenilin 1 (PS1)

PS1 is the catalytic core of the aspartyl protease γ-secretase, and its mutation is one of the major genetic causes of the familial form of AD (FAD). It is strongly expressed in parallel with Notch 1 in the ventricular zone of the embryonic brain in mice. However, the level of both proteins gradually decreases as the embryo develops, and they are weakly present in the neuroprogenitor cells and cerebellum during adulthood [50,51,52]. Indeed, PS1 is not only implicated in the modulation of brain plasticity in the embryonic brain but it is also a key regulator in Notch and Wnt signaling pathways. Consequently, PS1 has a crucial role in the developmental maturation of the glia and neurons [24,53].

#### 2.2.2. Tau

Tau is a highly soluble neuronal microtubule associated protein that plays a crucial role in the modulation of axonal microtubules [24,54]. It has a potential role in adult hippocampal neurogenesis, since its expression and post-translational modification are tightly regulated during the adult neurogenesis process [52,55,56,57]. Indeed, in a mouse model of tauopathy, neurogenesis was found to be severely impaired, especially due to a decrease in proliferation in the subgranular zone (SGZ) of the dentate gyrus (DG) and the subventricular zone (SVZ) in a cell autonomous fashion, and this was mainly attributed to the increase in the hyperphosphorylation of tau within the neural precursor cells per se [58]. Furthermore, the presence of an N-terminal pathological fragment of tau from 26–230 amino acids in AD patients was found to decrease the proliferation of neuroprogenitor cells and the long-term survival of newborn neurons in the adult SGZ [59]. In contrast, the ablation of tau was reported to cause an increase in the proliferation of neuroblasts and newborn neurons in the SGZ in mice [52,60,61]. Conversely, the depletion of adult neurogenesis in a familial AD (FAD) mouse model was found to cause an increase in the hyperphosphorylation of tau [62]. Hence, it is legitimate to postulate that there is a close relationship between the hyperphosphorylation of tau and defective neurogenesis in AD, and that decrease in tau hyperphosphorylation would be able to rescue the adult neurogenesis impairment in AD patients [52]. One possible mechanism through which tau can impact on adult hippocampal neurogenesis could be through its accumulation in the hyperphosphorylated form in GABAergic interneurons of the DG in both AD patients and in the 3xTg-AD transgenic mouse model of AD [63]. Consequently, the reduction of GABA and the disinhibition of local circuits and astrogliosis could in turn impair adult hippocampal neurogenesis [64].

GSK3-β is a kinase that hyperphosphorylates tau, and the increase in its activity is believed to be one of the causes of the AD pathology. GSK3-β causes an enhancement of the hyperphosphorylation of tau, and its inhibition was reported to promote neurogenesis in several studies in vivo as well as in vitro [65]. The hyperphosphorylation of tau caused by the activation of GSK-3β was also reported to result in the detachment of tau from the microtubules and inhibition of the insulin and Wnt/β-catenin signaling [52,66]. P021, a neurogenic neurotrophic compound which activates BDNF signaling, was found to cause an increase in neurogenesis, synaptic plasticity and a reduction in the hyperphosphorylation of tau via the tropomyosin-related kinase B (TrkB)/phosphoinositide 3-kinase (PI3K)/protein kinase B (Akt)-GSK3-β pathway in aged Fisher rats and 3xTg-AD mice [47,67,68,69].

#### 2.2.3. Amyloid-β

The sequential cleavage of APP by the two proteases, β-secretase and γ-secretase, generates the Aβ fragment (39–43 amino acids) which is more prone to aggregation and the formation of amyloid plaques [52]. Several studies reported a defective neurogenesis tightly linked to the deposition of Aβ. However, no direct evidence was found to prove this relationship. Furthermore, the extracellular deposition of Aβ was reported to decrease type 1 and type 2 progenitor cells in the APP/PS1 mice in the SGZ and the extraction of soluble Aβ from the hippocampus of this mouse model was able to block neuronal proliferation in vitro, probably through the activation of microglia [52,70]. Furthermore, Aβ deposition was also proposed to result in morphological and functional deficits in adult-born DG cells during later maturation by creating a situation of imbalance between gamma-aminobutyric acid (GABAergic) and glutamatergic inputs in AD [71]. Aβ could also bind to the p75 neurotrophin receptor on the membrane or primary cilia of nerve cells, thereby initiating apoptosis or decreasing the survival of neuroblasts [52,72].

In other neurodegenerative diseases such as Parkinson’s disease and Huntington’s disease, both key proteins: alpha-synuclein and Huntingtin, were proposed to play a crucial role in the plasticity of the brain [73]. Huntingtin was proposed to be in the cytoplasm and intervene in vesicular trafficking, transcriptional regulation, nucleo-plasmic shuttling and synaptic function [24,73]. Specific alterations to the neurogenic niche (DG, SVZ and/or olfactory bulb) are congruent with the early or premotor symptoms such as depression, anxiety or olfactory dysfunction that are seen in the early stages of these two neurodegenerative diseases [74]. Therefore, it is noteworthy that the mechanism of neurodegenerative diseases is strongly linked to deficits in brain plasticity [24]. Indeed, in neurodegenerative diseases, neuronal dysfunction appears at the level of synaptic transmission, synaptic contacts, and axonal and dendritic degeneration. Neurodegeneration in these diseases affects the specific population of neurotransmitters. In addition, an alteration of adult neurogenesis and the formation of new functional neurons were reported [24].

#### 2.2.4. APOE

APOE was reported to be primarily involved in the transport of lipids and cholesterol into neurons and mainly secreted by astrocytes. It also plays a crucial role in synaptogenesis, cerebrovascular integrity, cerebral blood flow, neuroimmune modulation, amyloid clearance, and hippocampal neurogenesis [75]. It is also expressed by adult neural stem cells, where it regulates their proliferative rate [76]. Three common isoforms: APOE 2, 3 and 4, are present in humans. The APOE 3 is the most common one with an allele frequency of 70–80% [77]. The APOE4 isoform is the one that is strongly linked to sporadic AD as a genetic risk factor; APOE 2 is believed to have a protective role and lowers the risk of developing AD [52]. APOE4 knock-in mice were reported to have an enhanced proliferation of neural stem cells but a reduction in the maturation of newborn neurons and less elaborate dendrites in the hippocampus, which may be due to the decrease in the number of GABAergic interneurons [78]. In addition, the facilitation of neurogenesis following environmental enrichment stimulation failed in the ApoE4-Tg mice, while it enhanced neurogenesis in the SGZ of ApoE3-Tg and wild-type mice [52,79].Therefore, the aforementioned studies show that the central molecular players in AD influence the generation of new hippocampal neurons, hence it is becoming evident that alterations in neurogenesis start even earlier than the onset of hallmark lesions or neuronal loss [22,80].

### 2.3. Neurogenesis in AD

#### 2.3.1. Neurogenesis

Adult neurogenesis in SGZ of the dentate gyrus of the hippocampus and SVZ are known to occur throughout life. Spalding et al. (2013) reported that the quantification of hippocampal neurogenesis in the adult human has a rate of 700 new neurons per day, and only 35% of the DG neurons undergo turnover with a renewal rate of 1.75% per year, leading to a full renewal of the neuronal population in the DG across the lifetime [20,81]. The newborn cells incorporate in the hippocampal circuit and play a crucial role in learning and memory. On the other hand, several studies emphasized the presence of impairment in neurogenesis in the AD brain, which probably contributes to the cognitive and memory impairment characteristic of the disease [20]. In addition, neurogenesis was reported to be defective in several other neurodegenerative diseases, such as Parkinson’s and Huntington’s diseases [24,43,82].

In mammals, adult neurogenesis occurs principally in two regions: the SGZ of the hippocampus and the SVZ (Figure 1). These two areas are thought to maintain a neurogenic stem cell niche [43]. The neural precursor cells that are found in both areas are a subset of astrocytes that give rise to immediate progenitors which migrate and differentiate into new neurons in the hippocampus (SGZ neurogenesis) or the olfactory bulb (in SVZ neurogenesis) [82]. In the SGZ, mature granular cells pass through several developmental stages. The proliferative stage, or stage 1, which is characterized by the presence of neural stem cells or type 1 radial glial-like cells that express glial fibrillary acidic protein (GFAP), nestin, and the sex-determining regionY-box 2 (Sox2); and the differentiation stage. Stage 2 is manifested by the presence of intermediate progenitor cells (type 2 cells) that express doublecortin (DCX) or polysialylated neural cell adhesion molecule (PSA-NCAM). Stage 3 is the migration stage, during which the type 2 cells give rise to the type 3 cells or neuroblasts, also called neuronal lineage; committed cells express DCX, PSA-NCAM, and markers for immature neurons (Tuj-1b or NeuroD). Stage 4 is characterized by axonal and dendritic targeting and the synaptic integration. Stage 5 is characterized by the presence of mature neurons that express calbindin [43,83]. Once matured, the newly formed DG cells differ completely from their neighbor older ones, especially in terms of electrophysiological properties. Neural stem cells that are found in the DG of the hippocampus primarily generate granular neurons that are excitatory and make up the bulk of the DG. In fact, they show enhanced synaptic plasticity with both increased amplitude and decreased induction threshold for LTP [80,84,85]. These newly formed DG cells integrate into the pre-existing neural circuits and become functional in the hippocampus after undergoing neurite remodeling for competitive horizontal-to-radial repositioning [52,86].

The entorhinal cortex (EC) sends input to the DG, which in turn sends axonal projections from the newly formed neurons through the tri-synaptic circuit to CA3 and CA1 of the hippocampus and dendrites into the outer molecular layer of the DG, forming synapses with neurons in layer II of the entorhinal cortex. In turn, CA1 projects to the subiculum and sends the hippocampal output back to the deep layers of EC, therefore playing a role in learning and memory [9,20,80]. In fact, several recent studies reported that the adult hippocampal neurogenesis has an integral role in learning and emotional regulation [24,43,87,88], and that it can contribute to brain plasticity in adulthood [43]. Adult hippocampal neurogenesis has several potential functions such as increasing resilience against stress, enhancing pattern separation [89], the formation of memory and learning [90,91] and inducing the loss of established or old memories [92]. Indeed, AD mouse model studies have shown that altered neurogenesis was linked to deficits in pattern integration, pattern separation, and cognitive flexibility [64]. In a study conducted by Nakashiba et al., it was reported that older neurons are mainly implicated in pattern completion as they recognize relatively distinct situations, while younger neurons are essential for the fine discrimination of close contexts. Therefore, this study suggests age-related functional changes of neurons and that continuous neurogenesis in adults is crucial for memory discrimination [20,93]. Consequently, adult hippocampal neurogenesis has a crucial role in hippocampal plasticity [89,94] and remodeling of the hippocampal circuits [43]. A regulated and appropriate level of neurogenesis is thus needed for the hippocampus to balance old memory storage and new memory formation. Additionally, a timely clearance of old memories will improve the efficiency in developing and storing recent memories within the already existing hippocampal network [20,95]. Since the hippocampus and EC are two key regions involved in memory and they are particularly vulnerable to neurodegeneration and are the first affected during the AD pathogenesis, deficits in neurogenesis probably play a significant role in this disease [64].

#### 2.3.2. Neurogenesis in AD

It is very well documented that neurogenesis decreases with age both in the SVZ and the SGZ. Neurogenesis was found to be reduced with aging in both regions, with a severe loss of proliferation in rodents [9,96], non-human primates [97,98], and in humans [99,100]. Furthermore, alterations in adult hippocampal neurogenesis are considered to begin several years before clinical manifestation in AD patients, even before the deposition of Aβ, formation of neurofibrillary tangles and inflammation [52]. Hence, adult hippocampal neurogenesis could be considered as an integral part of AD pathology [80].

Recent studies postulated that the reduction in neurogenesis that occurs in physiological aging is worsened in AD [9]. However, results were contradictory with some reporting an increase and others a decrease in adult hippocampal neurogenesis in AD patients and AD mouse models. A research group reported that neurogenesis was increased in human AD patients, which was explained by the presence of a self-compensatory mechanism to replace the degenerated neurons [80,101]. However, the number of these newly born neurons is small and consequently they are unlikely to replace the huge number of degenerating neurons in AD. Therefore, the authors suggested that this compensatory mechanism could slow down the cognitive decline, but it would fail in achieving a global repair [80]. They further hypothesized that the newly produced neurons could be non-functional because they cannot develop into fully mature neurons or to the right type of neurons, or they are unable to integrate into the surviving brain circuitry [101]. The answer came from a pioneering study from our lab where we showed that the expression of the mature neuronal marker, high molecular weight microtubule-associated protein (MAP) isoforms MAP2 a and b was severely reduced in AD patients in the DG (1%) as determined by immunohistochemistry and in situ hybridization. Furthermore, we showed that the total MAP2, including expression of the immature neuronal marker, the MAP2c isoform, was less affected (60%). This suggests that MAP2c is mainly expressed in the DG of AD patients. From these findings it became clear that there is a failure in the process of maturation of the newly born neurons in the DG, although neural proliferation is increased [15]. Hence, an arrest in maturation of the developmentally immature granule cells could have happened because of the changes to the brain microenvironment at the beginning of the disease, especially the marked imbalance of neurotrophic factors, including the increased level of fibroblast growth factor 2 (FGF−2) and decreased levels of the brain-derived neurotrophic factor (BDNF) and neurotrophin 4 [15,102,103,104] (Figure 1).

Three independent studies reported a sharp reduction in the adult hippocampal neurogenesis in the AD brain. Masliah and colleagues reported a huge reduction in DCX+ and Sox2+ cells in the dentate gyrus of AD hippocampus as compared with nondemented control cases [43,105]. Moreno-Jimenez et al. quantified the DCX−positive immature neurons in 13 healthy patients at the ages of 43 and 87 years and 45 AD patients between the age of 52 and 97 years of age, and they found that the number of DCX+ cells was markedly reduced in AD patients compared to controls, especially with the advancement of the disease and that maturation of the newly born neurons was defective [43,106]. Tobin et al. employed three study groups: healthy aging, mild cognitive impairment (MCI), and AD, and they found that adult hippocampal neurogenesis still exists till the 10th decade of life in healthy humans. In addition, they reported the detection of neural stem cells, neural progenitor cells and new neurons in MCI and AD patients, which correlated with their clinical diagnosis, indicating that adult hippocampal neurogenesis is affected early in the disease process of AD and could be a potential target for early intervention [99].

Although causal links are not yet fully clarified, dysfunction in adult hippocampal neurogenesis can at least partially be responsible for the cognitive deficits and hippocampal atrophy in AD [52]. Hence, alteration in neurogenesis resulting from early disease manifestations may in turn worsen neuronal vulnerability to AD and contribute to memory impairment [20]. In fact, impaired neurogenesis may interfere with synaptic and neuronal plasticity as well as normal neuronal function [47]. However, enhanced neurogenesis could be a compensatory mechanism and represent an attempt by the brain to self-repair [80].

### 2.4. Neurodegeneration, Synaptic Deficit, and Synaptic Compensation

#### 2.4.1. Neurodegeneration

In AD, neurodegeneration is an established process. In normal aged individuals, the loss of brain mass/year could reach ~0.5% and in AD it is ~5-fold higher [107]. Brain mass loss is estimated to be 200–400 g after progression of the disease for a period of 7–10 years [108]. The hippocampus is a major region of neuronal loss and atrophy reaching~10% per year. Neuronal loss in AD reaches more than 50% of the neurons and the loss increases with the progression of the disease. Neurodegeneration in the cortical association areas has been directly linked to memory loss [33]. Neuronal loss is most prominent in layer II of the entorhinal cortex in mild AD, which distinguishes it from that layer in non-demented aged individuals. In addition, neurodegeneration was found to correlate mostly with the density of NFT and not of senile plaques [11]. Indeed, a major contributor to neuronal loss could be the hyperphosphorylation of tau since it sequesters normal tau, which causes the breakdown of microtubules. This would compromise axonal transport and lead to retrograde degeneration and synaptic loss [109]. These events are believed to lead to brain volume loss and dementia. Another key mechanism could be the imbalance of neurotrophic factors. A small change in the NGF level in transgenic mice is linked to impaired synaptic plasticity and cognitive performance. Furthermore, in AD the BDNF polymorphism from Valine to Methionine occurs with high frequency, and it has been shown to inhibit the cleavage of Pro-BDNF to BDNF [110].

#### 2.4.2. Synaptic Deficit

Synaptic plasticity is a very important process in the development and function of neuronal networks, and hence it is considered a cellular substrate of learning and memory [111]. In fact, the very early stages of AD were suggested to start with synaptic deficit followed by neurodegeneration and then Aβ and tau pathologies and cognitive impairment [47,112]. Synaptic loss is used as a consistent feature to differentiate between the brains of demented and non-demented people and is directly linked to the severity of dementia [113,114,115]. Synaptic loss correlates well with other types of dementia and age-associated decrease in cognitive performance, called normal cognitive aging [47,110] (Figure 2).

AD is characterized as a synaptic failure [112,116] with a loss of dendrites and dendritic spines [117]. In 2−4 years after the clinical diagnosis of AD, quantitative evaluation of AD brains showed that there was a reduction of 25–35% in the number of synapses per neuron in the frontal and temporal cortices [118]. This loss is more aggravated in the hippocampus, where it reaches 44–55% [113,115,119]. Synaptic loss in the frontal cortex and limbic regions directly correlates with the severity of cognitive impairment, especially the decrease in the presynaptic marker, synaptophysin, and with the increase in the number of NFT in AD [113,114]. In fact, synaptic loss is thought to start first in the EC with dendritic neurodegeneration, which makes up to 90% of the contacts. Synaptic loss is not limited to only the tangle-bearing neurons but also the non-tangle-bearing neurons and the major synaptic loss happens in the early stages of the disease [118,120]. The ratio of synapses to neurons drops by 48% [121]. Because synaptic loss in the surviving neurons accounts for 38%; cognitive impairment is believed to happen not only because of the synaptic loss but also because of the impaired capacity of the still surviving synapses [122]. Neuronal and synaptic plasticity are known to be key modulators of neuronal firing, neuronal recruitment into information processing networks, and ultimately learning and memory mechanisms. Therefore, the synaptic deficit in AD might not only involve direct damage to the synapses, but also interference with neurogenesis [123].

#### 2.4.3. Synaptic Compensation

Another phenomenon that was reported very early in the disease process in MCI patients and rodents is a transient increase in the level of synaptic markers to compensate for synaptic loss, as was seen during the deficit in neurogenesis. Several reports showed that the levels of synaptophysin and other synaptic proteins were enhanced very early in the disease process of AD prior to NFT formation (BRAAK stage III) and then decreased again when the disease progressed, suggesting a phenomenon of synaptic compensation [119].

Furthermore, an enhancement in the level of PSD-95 in AD patients [124,125] and the presynaptic cholinergic bouton density in the midfrontal gyrus of MCI patients was reported [126]. An increase in the synaptic size was seen accompanying synaptic loss in AD to compensate for it. Indeed, in several neocortical areas the increase in synaptic size compensates for the significant loss of synapses by maintaining a constant total synaptic contact area. However, when the disease progresses and synaptic loss becomes more intense in these areas, the compensation phenomenon fails. This failure is more clear in the neocortical areas that are known to be implicated early in synaptic loss such as Brodmann area 9 [119]. fMRI studies corroborate these findings, which may suggest the presence of a biphasic stage in the prodromal stage of the disease in which there is a period of increased brain activation which is reduced later during the disease process [127]. Consequently, the compensation phenomenon may be effective in the early stages of the disease, which slows down its progression but then becomes defective at the advanced stages and the progression of the disease may even be accelerated [47,128].

In AD, synaptic and neuronal dysfunction is probably caused by a combination of different etiopathogenic factors including the accumulation of hyperphosphorylated tau and Aβ, age-related processes, and neuroinflammation either by pro- or anti-neurogenic effects [14].

### 2.5. Neuroinflammation

Systemic inflammation or neuroinflammation, which is an early, special immune response to tissue damage or pathogen and affects the process of adult hippocampal neurogenesis either by pro-neurogenic or anti-neurogenic effects, is another feature of AD and other neurodegenerative diseases [24,43,82,129]. Whether there is an inhibition or an excitation depends on the activation of microglia, macrophages, or astrocytes and the duration of inflammation [43,82,130]. The balance between the benefits and the negative effects of inflammation could have a profound impact on the efficiency of brain repair [131], which is very important in the context of the neurodegenerative disorders [82]. Three main effector cells intervene in the process of neuroinflammation in the brain: mast cells, which attract and activate by secreting pro-inflammatory cytokines and chemoattractants [132]; astrocytes, which liberate both pro- and anti-inflammatory cytokines, chemokines and complement components [133]; and microglia, which are the main effector cells of the immune system and are found in a “resting” quiescent stage during normal physiological functions [52,82] (Figure 2). There is increasing evidence that microglia play a crucial role named as the pro-neurogenic effect in the regulation of adult hippocampal neurogenesis [52]. Indeed, microglia are reported to be responsible for the phagocytosis phenomenon in the process of adult hippocampal neurogenesis, therefore, the homeostatic maintenance of the neurogenic niche [134]. By the secretion of neurotrophic factors such as insulin-like growth factor 1 and trypsinogen, microglia seem to have an instructive effect on neural stem cell proliferation and differentiation, hence, regulating their fate [52,135,136,137]. It is estimated that between 30% and 40% of neural progenitor cells and neuroblasts contact microglial cells in the hippocampal dentate gyrus. Cells that are destined to apoptosis and are removed by microglia mainly represent young cells that are in the intermediate stage between late amplifying neuroprogenitor cells and early neuroblasts, or young postmitotic newborn neurons [134]. Thus, microglia play an important role in the regulation of the neurogenic niche ensuring proper survival, differentiation, and integration of newborn neurons, but in the meantime removing cellular components or degenerated neurons which could induce an inflammatory signaling cascade like that reported in AD [134,138,139]. The function and the activity of the microglia were recently reported to be regulated by the neuroprogenitor cells. Hence, factors that are secreted from the neuroprogenitor cells may modulate microglia activation, proliferation, and phagocytosis [82,140]. Evidence of the pro-neurogenic effect of microglia was reported using in vitro studies where microglia population in the resting state were found to release factors that rescue neuroblasts and instruct neuronal cell differentiation [82,141]. In AD, the overactivation of the microglia results in the shift in the microglial phenotype (especially the activated pro-inflammatory M1 phenotype) and the change in their morphology by retracting their fine processes and acquiring a reactive-like morphology [139]. Consequently, this results in the induction of the anti-neurogenic effect, hence halting adult neurogenesis by decreasing the survival of neuroblasts [43,52]. The activated microglia in AD secrete cytokines that inhibit adult hippocampal neurogenesis such as interleukin (IL)-6, tumor necrosis factor (TNF)-α, and IL-1β [134,138,139,142,143]. Specific receptors of these proinflammatory molecules are also activated, such as IL-1β receptor and TNF-α receptors 1 and 2 [144]. Furthermore, the activation of TNF-α and IL-6 would result in mitochondrial dysfunction during brain inflammation, causing a reduction in the energy supply necessary for adult hippocampal neurogenesis [145,146]. These reactivated microglia act also in synergy with astrocytes to enhance the inflammatory reaction and release more cytokines and chemokines to further increase the chemoattraction of microglia and macrophages. This results mainly in the increase in the permeability of the blood-brain barrier and the restriction of brain injury due to the gliotic reaction primarily driven by astrocytes [139]. Although direct evidence of an association between the deficit in neurogenesis and neuroinflammation is missing, the negative effect of neuroinflammation on adult hippocampal neurons was reported in several rodent studies [52].

Under pathogenic conditions, tau is hyperphosphorylated and unable to bind to microtubules, resulting in the formation of protein aggregates. Microglia and astrocytes are reported to be abundantly found near neurons and plaques, and there is an induction in the release of cytokines [82]. Therefore, the chronic inflammation seen in AD could be a response to the accumulation of A𝛽 plaques and tangles [147]. This chronic activation of microglia and astrocytes could result in the induction of necrosis of neighboring neurons by releasing reactive oxygen species, proteolytic enzymes, complementary factors, or excitatory amino acids [148]. A𝛽 and APP were also reported to strongly activate glia cells through the binding to the microglial cell surface, regulating extracellular signal regulated kinase (ERK) and mitogen-activated protein kinase (MAPK) pathways that enhance proinflammatory gene expression leading to cytokine and chemokine production [82,149].

Since neuroinflammation is an important feature of the disease and has a great impact on adult hippocampal neurogenesis, the modulation of the inflammatory environment could be beneficial not only to improve the deficits created by the disease but also to boost the brain self-repair mechanism against internal damage. Hence, in this context a deeper understanding of the role of adult hippocampal neurogenesis in AD is of major importance since the hippocampus is one of the neurogenic zones and it is the region that is mostly affected in AD and is mainly responsible for cognitive and learning capacities which are largely impaired in AD patients [82].

The final common outcome of all ethiopathogenic mechanisms in AD is the process of neurodegeneration leading to cognitive impairment.

### 2.6. Cognitive Impairment

AD interferes with the process of memory formation from the molecular level to the level of the whole brain network. In most AD cases, slow cognitive deterioration happens several years before overt clinical symptoms. Short-term memory is the first to be affected very early in the process of the disease, and patients develop difficulties in remembering new information and familiar people’s names. Furthermore, executive functions like judgment and problem solving and organizational skills are also affected; long-term and declarative memories are, however, less affected until the final stages of the disease. Small personality changes and behavioral changes are also reported. Aggression, anxiety, and psychosis are not reported in the initial stages. During moderate to severe stages, however, cognitive dysfunction becomes more pronounced and newly learned information is rapidly lost with patients described as “living in the past”. Spatial navigation difficulties and disorientation in the familiar places become marked symptoms as well as the failure to recognize family members and close relatives. Logical reasoning and executive functions are severely affected and behavioral symptoms such as hallucinations, delusions and illusionary misidentification become more common. Disruption in sleep patterns, agitation with temper tantrums, physical and verbal aggression and anxiety are also common in this stage. Apathy, however, is the most persistent and frequent symptom exhibited by 72% of the patients through all stages of AD [150]. At the final stages of the disease, almost all cognitive functions are disrupted and the patients become bedridden and totally dependent on caregivers [47] (Table 2).

### 2.7. Growth Factors and Neurotrophins

Growth factors such as vascular endothelial growth factor (VEGF), BDNF, insulin growth factor-1 *IGF-1), FGF-2, IGF and neurotrophins are known to play an extrinsic modulator role in neurogenesis and contribute to the proliferation, migration, cell fate determination and maturation of neural stem cells and neuroprogenitor cells [43,80,158,159,160,161] (Figure 2). In the AD hippocampus, a marked imbalance of these factors including an increase in the level of FGF-2 was found [15,104]. Indeed, the increase in the level of FGF-2 favors cell division and the level of nestin and reduces the level of neuronal lineage markers such as Tuj1 and MAP2a and b in adult hippocampal progenitor cells in culture [15,104]. This indicates that the elevated level of FGF-2 drives the cells to stay in an undifferentiated, actively dividing developmental stage [15]. Furthermore, in the aged hippocampus, levels of key neurotrophic factors, such as FGF-2, IGF-1 and VEGF were reported to be reduced [162]. Neurotrophic and growth factors such as BDNF and VEGF are also implicated in the proliferation, differentiation and integration of new neurons into the existing circuitry [162]. NGF was reported to intervene in various stages of neuronal precursor maturation in the SVZ. It is noteworthy that the age-dependent decrease in neurogenesis was correlated with the age-dependent decrease of NGF and other growth and hormonal factors [163]. NGF acts by tyrosine kinase (TrkA) and p75 receptors. Besides, the NGF-TrKA-PI3K-Akt pathway was shown to be essential for axonal growth and neuronal survival [164,165]. The PI3K/Akt, Wnt/β-catenin, and BDNF/TrkB/CREB signaling pathways are very critical in the regulation of adult hippocampal neurogenesis and are very likely to be responsible for the deficit in neurogenesis seen in AD [52]. Hence, the deficit in neurogenesis in AD could be due to a lack of neurotrophic support, and modulation of the neurotrophic environment may have therapeutic potential in brain regeneration [164,165].

We have mentioned above that neuroproliferation was found to be increased in DG, but these newly formed neurons failed to mature in AD [15]. Neurotrophic factors have been shown to prevent endogenous and experimentally induced neuronal cell death and promote and maintain neural stem cell differentiation and maturation [166,167,168]. The selection of the newly born neurons that survive depends on the availability of neurotrophic factors in the target neurons. Neurons release these neurotrophic factors in an activity-dependent manner. Therefore, a neuron that is not very well connected to its target neuron will die, since it does not receive sufficient neurotrophic support, and the very well-connected neuron will receive a significant amount and therefore survive to become a mature neuron [28]. Since synaptic loss and synaptic dysfunction are early events in the pathogenesis of AD, a loss of synapse results in a loss of connection between the innervating neuron and the target neuron and the synaptic transport dysfunction to defect the retrograde transport of the neurotrophic factors. These two phenomena could lead to the death of the innervating neurons due to lack of reception of neurotrophic factors released from their target counterpart [47].

Ciliary neurotrophic factor (CNTF), a 22.7 KDa protein, has a favored position since it has neuroprotective capacities [167,169,170]. It plays a pivotal role in adult hippocampal neurogenesis and SVZ neurogenesis and the differentiation of neural stem cells [166,171,172]. Signaling of CNTF occurs through a tripartite complex receptor α (CNTFRα), LIF β receptor (LIFR), and glycoprotein 130 (gp130). CNTF and leukemia inhibitory factor (LIF) both signal via tyrosine phosphorylation (Tyr706) of the signal transducers and activators of transcription (STAT) proteins by the membrane associated Janus kinase (JAK) [173]. CNTF is expressed by astrocytes in the neurogenic niche of the brain and its receptor, CNTFRα, predominantly by neural progenitor cells and hippocampal neurons, and various other areas of the brain such as the motor cortex and cerebellum [166,173]. The administration of the full-length protein of CNTF in human clinical trials generated several side effects such as anorexia, muscle pain, gastrointestinal symptoms, weight loss and analgesia [47]. Furthermore, its use was halted by the limited blood–brain barrier (BBB) permeability, poor plasma stability, unsuitable pharmacokinetics, and unwanted systemic effects [174,175,176,177,178]. Hence, to benefit from these neuroprotective characteristics of the CNTF but minimizing the side effects of the administration of the full-length protein, we have focused in our lab on developing a small peptide mimetic through epitope mapping that can boost neurogenesis and neuronal plasticity and, consequently, enhance cognitive function without the deleterious effects of the full length parent protein. These small molecule mimetics could modulate various aspects of the signaling pathways in a way distinct from the conventional neurotrophic factor signaling. Thus, these small-molecule mimetics might provide a novel therapeutic approach to treat AD [174,179]. Since these molecules are small, they offer several advantages compared to native neurotrophic factors, such as suitable pharmacokinetics and enhanced BBB permeability. However, insufficient receptor specificity, requirement of continuous doses and effects which are not brain region-specific [179] should be taken into consideration [174].

## 3. Development of Drugs for AD

### 3.1. Shift from Large Molecules to Small Peptidergic Compounds

#### 3.1.1. Peptides as Drugs

Proteins are large polypeptide chains containing 100 amino acids and up, while peptides are small molecules formed through a peptide bond between two or more amino acid residues [180]. These latter are considered efficient as a new drug source since they are highly selective, efficacious, and weakly toxic [180]. These magnificent properties increase their potential to be used in disease modifying therapies [181]. Despite the presence of problems of short-half life in vivo and limited brain bioavailability, these obstacles can be overcome with a rational drug design [182]. Drugs that are now present in the market have a relatively small size, are lipohilic and can cross the BBB [183]. These peptide drugs are more efficient than antibodies that are used in passive and active immunization therapy since they are big and have very limited BBB permeability [183]. For most central nervous system (CNS) drugs on the market, less than 0.2% of the peripheral dose is taken up by the brain [183].

The major difficulties facing peptide drug delivery are the presence of the endothelial cell wall as a physical barrier, binding to proteins in the circulation and degradation by enzymes in the circulation at the BBB or within the CNS, uptake or sequestration by peripheral tissues of the peripherally administered substance, sequestration by the capillaries that comprise the BBB and efflux, or removal by CNS-to-blood transporters. Taking these unfavorable pharmacokinetics into account while designing a drug would make it more successful in the market [184]. The chemical manipulability of the peptides by their conjugation with other small molecules or the incorporation of non-natural amino acids by design can help in overcoming these difficulties. In fact, the addition of non-natural amino acids can also prevent rapid degradation of the peptide by rendering the peptide unrecognizable to proteases, hence decreasing its degradation and increasing its half-life in the plasma [184]. The use of peptide drugs as molecules open up a diversity of drug route delivery options other than intravenous injection, improve patient compliance and reduce the overall cost of treatment [184]. Several synthetic as well as natural peptides, such as carnosine, are being developed to treat several neurodegenerative diseases. Some crucial peptide inhibitors that are currently in use for neurodegenerative disorders include Aβ (16–20) KLVFF for Alzheimer’s disease, NAPVSIPQ (NAP) and DNL201 (an LRRK2 inhibitor) for Parkinson’s disease [185], Vasoactive Intestinal Peptides (VIP) for Huntington’s disease, and Polyglutamine Binding Peptide-1(QBP1) for Dentatorubral-pallidoluysian atrophy (DRPLA) [181]. Peptide mimetics that are derived from neurotrophic factors are considered as very promising drugs for neurodegenerative diseases, especially AD.

#### 3.1.2. Neurotrophic Factor Peptide Mimetics as Potential Drugs for AD

Several therapeutic strategies were developed for AD. However, so far, none of them have resulted in an effective treatment or have been shown to even slow down AD. A major focus was placed on the development of drugs that target amyloid β synthesis and accumulation. The repeated failures of the anti-amyloid therapies in clinical trials and the severe side effects are increasingly shifting the focus to anti-tau therapies. After all, the density of tau pathology and not Aβ pathology correlates with dementia [12]. Tau-based therapeutic approaches include the inhibition of tau hyper-phosphorylation and other posttranslational modifications, aggregation, the promotion of tau clearance and the prevention of prion-like spread of tau pathology [14,186]. The anti-tau therapies field is still nascent, and it will take the next several years to learn whether targeting this single lesion can lead to the effective treatment of AD.

AD is a multifactorial disease characterized by progressive neurodegeneration. Probably because of the age and the underlying pathology, the AD brain lacks the sufficient appropriate neurotrophic support to successfully regenerate and rescue the lost connectivity. Thus, a rational therapeutic approach that can effectively prevent and treat AD is by using a neurogenic/neurotrophic compound that can shift the balance from neurodegeneration to neural regeneration. This can probably be achieved with a neurotrophic small peptide mimetic that can cross the BBB, weakly degraded by peptidases, and act on a target that can lead to neural regeneration and rescue connectivity in the affected areas of the brain. In ligand mimetics, large polypeptides are reduced to small functional units that contain the site of interaction in which the molecule can modulate specific receptors. What makes mimetics useful is that the protein-protein interaction occurs only in a few key regions or “hot-spots” instead of the overall protein surface. Two approaches have been used to develop small peptide mimetics. The first one is to mimic antibodies for neurotrophin receptors, and the second one is to mimic the ligand itself (neurotrophins). Both antibodies and neurotrophin mimetics should bind to the same receptor as the original molecule from which they are derived [47,187].

In our lab we have developed a promising therapeutic strategy that mainly focuses on shifting the balance from neurodegeneration to regeneration of the brain with neurotrophic compounds to help the brain’s attempt to self-repair by enhancing neurogenesis, neuronal plasticity, and reducing the accumulation of the two pathological hallmarks of AD (Aβ plaques and NFT) and prevent cognitive impairment starting very early in the disease process. Since neuronal and synaptic loss play a crucial role in cognitive and memory impairment, one would postulate that a small molecule that has a dual function of promoting neurogenesis and neuroprotection could inhibit cognitive decline and reverse the disease state [188]. Increasing adult hippocampal neurogenesis and stimulating neuronal plasticity pharmacologically is considered a very useful strategy towards inhibiting cognitive decline in AD [47]. The brain in AD responds to neurodegeneration by stimulating neurogenesis and neuronal plasticity in the hippocampus, which were unsuccessful probably because of the lack of the appropriate neurotrophic support [15,128]. Thus, one potential rational therapeutic approach to AD and other neurodegenerative conditions is to provide a neurotrophic environment in the brain that can materialize into successful neurogenesis and rescue the neuronal plasticity deficit [189].

In our lab, we have developed a neurogenic/neurotrophic compound named P021 (Figure 3). P021 is a neurotrophic peptidergic compound that was derived from the active region of the CNTF, the amino acid residues 147 to 150 through epitope mapping of neutralizing antibodies [104,174,190]. P021 is adamantylated on its C-terminal to improve its lipophilicity, increase its BBB permeability, and decrease its degradation by exopeptidases [191]. P021 has favorable pharmacokinetics for drug development since it has a plasma half-life of >3 h and stability of >90% and ~100% in artificial gastric and intestinal fluids at 37 degrees centigrade for 30 min and 120 min, respectively, and is BBB permeable [47,68,192]. Up to 18 months of oral administration of P021 did not induce any adverse effects in 3xTg-AD or in wild type control mice [68,174].

P021 acts by modulating the CNTF pathway through the inhibition of the anti-neurogenic activity of leukemia inhibitory factor [193] and the increase in the expression of the BDNF, thus increasing the survival, maturation, and integration of newborn cells, thereby boosting neurogenesis [174]. P021 robustly inhibits GSK-3β activity, mainly through the increase in BDNF expression [174]. BDNF activates PI3K-AKT signaling that results in the downstream inhibition of GSK3β by inducing its phosphorylation at Ser-9 by AKT [123]. Since GSK3β is a major tau Serine/threonine kinase that phosphorylates tau at many different sites including Ser199, Ser202, Thr205, Ser396, and Ser404, its inhibition decreases tau’s hyperphosphorylation and the amyloidogenic processing of AβPP [122,194,195]. The hyperactivation of GSK3β was also reported to impair neurogenesis and initiate apoptosis. Consequently, its inhibition would rescue neurogenesis and reduce neurodegeneration [47,196]. This would result in a reduction of memory and cognitive deficits seen in AD. Hence, this small peptide mimetic helps in overcoming the main limitations associated with the therapeutic usage of neurotrophic factors such as CNTF and BDNF, since when peripherally administrated they are ineffective at reaching the CNS and are degraded within a few minutes [197].

Using several neurodegenerative mouse models in pre-clinical studies, we reported that P021 was effective as a disease modifying compound in advanced disease stages and in the prevention of AD and Down’s syndrome (DS). In a previous study using C57Bl/6 mice, we showed that peptide 6 (P6), an 11-mer peptide, a parent molecule of P021 and a subsequence of it, peptide 6c, enhanced hippocampus-dependent learning and memory, enhanced neurogenesis and neuronal plasticity in normal adult mice [167,190]. The intraperitoneal injection of P6 for six weeks in 6–7-month-old 3xTg-AD mice at early disease stage before any overt Aβ or tau pathologies rescued impairment in spatial reference memory and short-term episodic memory by inducing neurogenesis and neuronal plasticity in these mice [174,198,199]. In preclinical studies using animal models of sporadic AD [200], familial AD [201], Down’s syndrome [111,193], autism [202], traumatic brain injury [190], and cognitive aging [67] P6 was also shown to have neurogenic and neurotrophic effects by increasing neurogenesis, synaptic plasticity and inhibiting cognitive deterioration [174]. P021, which was derived from P6, was also found to enhance neurogenesis and synaptic plasticity via the increase in BDNF expression and by decreasing tau levels in aged Fisher rats [67,174,203]. Peripheral administration of P021 enhanced learning as well as both short-term and spatial reference memories of normal adult C57Bl6 mice [191]. It also has a disease modifying effect in 3xTg-AD mice aged 9–10 months when treated for 12 months at 60 nmol/g feed. It reduced Aβ generation, the synaptic and neurogenesis deficit and improved cognitive impairment at moderate to severe stages of the disease [68]. The administration of P021 orally to mouse mothers during gestation and weaning of their offspring from prenatal day 8 to postnatal day 21 resulted in an increase in several synaptic markers including NR1, NR2A, and CREB expression, rescue of the PSD95 deficit and prevention of tau and Aβ pathologies [197]. In a subsequent secondary prevention study, P021 at 60 nmol/g feed in 3xTg-AD mice at the synaptic compensation stage starting from the age of 3 months up to 22 months of age was able to completely prevent synaptic and neurogenesis deficits, reduce neurodegeneration, and prevent cognitive impairment as well as Aβ and tau pathologies [204]. The use of P021 even reduced the mortality level in these mice by almost 50% [204].

Age associated macular neurodegeneration (AMD) is a known co-morbidity of AD and is also associated with cognitive aging. The chronic treatment with P021 was found to prevent AMD in aged rats and 3xTg-AD mice [205]. In a recent primary prevention study, prenatal to early postnatal treatment with P021 was found to prevent cognitive deficits, AD-type pathological changes, which include tau and Aβ pathologies, postsynaptic deficit, and neuroinflammation later in life in 3xTg mice [197]. Neuroinflammation prevention could be by virtue of its effect on Aβ and tau pathologies. Similarly, treatment with its parent molecule, P6, in autism sera treated rats produced a reduction in astrogliosis [202]. Reduction of neuroinflammation could in turn contribute to a beneficial effect of P021 on neuronal and synaptic deficits and cognition [197].

#### 3.1.3. Current Status of AD Approved and Developing Drugs in the Market

Like most of the neurodegenerative diseases, therapeutic approaches of AD are divided into three categories: symptomatic, disease modifying (DMT) and regenerative therapies. Despite the approval of the effect on cognitive function that some approved drugs have, a more significant unmet medical need for symptomatic treatments is required to have a stronger effect on the cognitive domains and other distressing symptoms such as agitation, psychosis, and sleep disturbance [206]. Most of the theories that are being developed currently focus mainly on the disease modifying/disease progression theories in hopes of delaying the disease onset or to slow it down, hence people will live longer without developing the disease [206]. The currently approved drugs are cholinesterase inhibitors, agonists of the cholinergic system, (rivastigmine, donepezil and galantamine) and an antagonist of the N-methyl-D-aspartate receptor (Memantine). The drug Memantine was approved for moderate to severe stages, and it showed some positive effects on patient cognition, function and associated social benefits. The impact, however, on the patient quality of life was inconclusive (Table 3).

Historically, the drug development for AD has shown a remarkably high failure rate. Indeed, only one out of 244 from 2002–2012, only memantine has completed the third trial with a success rate of 0.4%. In a detailed and comprehensive review by Cummings and colleagues (2022), it was reported that during the year 2022 there were 31 agents in phase 3 clinical trials. Twenty-one of them are DMTs, five are biologic and 16 are small molecules (Table 3). In phase 2 clinical trials, there are 82 agents, 71 of them are DMTs, 26 are biologic and 45 are small molecules. In a phase 1 clinical trial there are 30 agents, 27 are DMTs, nine of them are biologic and 18 are small molecules (Table 3). However, the biggest question that remains is whether any drug will complete the phase 3 clinical trials and whether the mechanisms of the disease were adequately tested.

## 4. Conclusions

AD is a complex multifactorial disease for which, at present, no effective drug to prevent or treat the disease is available. The repeated failures of several recent clinical trials, especially those targeting Aβ, make us realize that focusing only on one component of the disease is not likely to be effective. Instead, using a combination therapy or small peptide mimetics that work on more than one molecular target acting mainly to enhance the plasticity of the brain and help the brain in its self repair attempt could be effective for AD. In the present review, we have described the complex nature, the reasons for lack of effective treatments, and a rationale and a promising therapeutic approach for the treatment of AD and related neurodegenerative conditions with a neurotrophic peptidergic compound. Nevertheless, it is important to note that the precise timing of intervention as well as the stratification of patients into subgroups could have a major impact on the design of the drug trial. Indeed, from the plethora of available data in the literature we realize that treatment must start very early in the disease process during the synaptic and neuronal compensation period when the brain still has the capacity of self-repair. In fact, once the disease reaches a certain point, neurodegeneration becomes excessive and irreversible and aberrant neural networks cannot be repaired by the simple reduction of Aβ and tau pathologies.

## Figures and Tables

**Figure 1 biomolecules-12-01409-f001:**
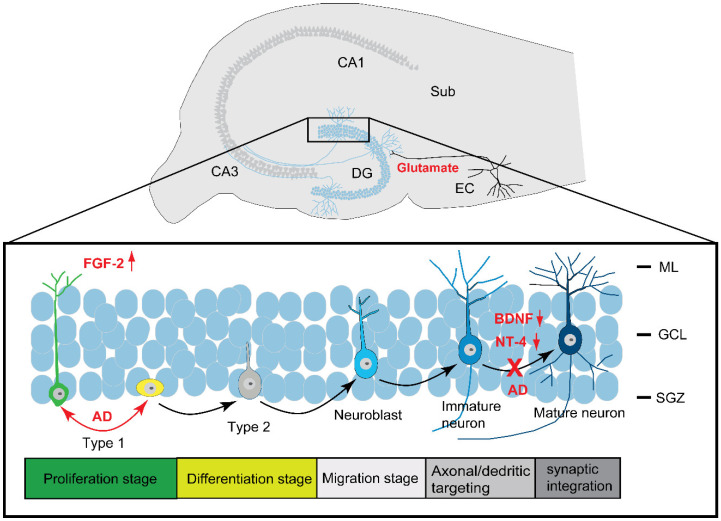
Neurogenesis in AD. The proliferative stage, or stage 1, is characterized by the formation of type 1 radial glial-like cells that express glial fibrillary acidic protein (GFAP), nestin, and sex-determining region Y-box 2 (Sox2). The differentiation stage, or stage 2, is characterized by the presence of intermediate progenitor cells (type 2 cells) that express doublecortin (DCX) or polysialylated neural cell adhesion molecules (PSA-NCAM). During the migration stage, or stage 3, the type 2 cells give rise to the type 3 cells or neuroblasts also called neuronal lineage committed cells that express DCX, PSA-NCAM, and markers for immature neurons (Tuj-1b or NeuroD). The axonal and dendritic targeting or stage 4 and the synaptic integration or stage 5 are characterized by the presence of mature neurons that express calbindin. The Alzheimer’s pathology-associated proteins such as hyperphosphorylated tau, PSEN 1, 2 mutations, Aβ, the APOE4 isoforms and the tau kinase GSK3β are known to affect the process of neurogenesis in AD at different stages. The imbalance of growth factors is also an essential cause of a defective neurogenesis in AD.

**Figure 2 biomolecules-12-01409-f002:**
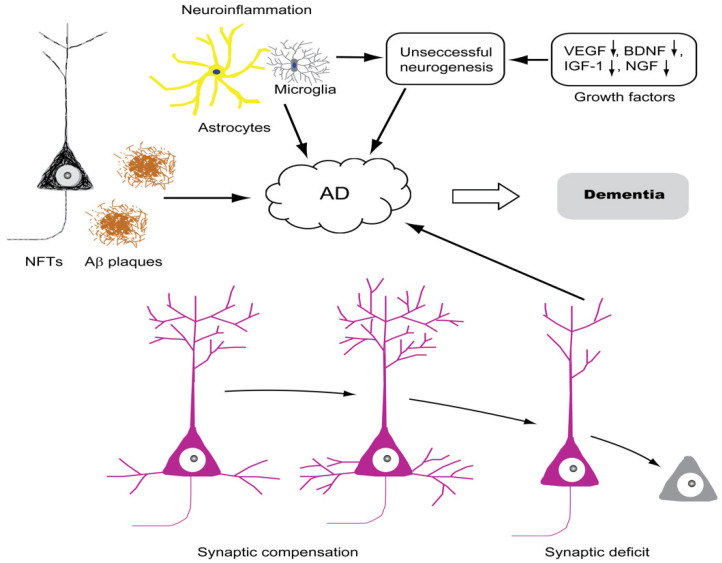
AD is a multifactorial disease. AD is a proteopathy that is characterized by the extracellular deposition of Aβ plaques and the intracellular deposition of NFT of hyperphosphorylated tau. Neurodegeneration is an established process in AD and its presence in the cortical association areas directly correlates with memory loss. A transient increase in the level of synaptic markers known as the phenomenon of synaptic compensation is present at the early stages of the disease, which is followed by the loss of synapses and synaptic deficit. Synaptic deficit also directly correlates with cognitive impairment in AD patients. Systemic inflammation and cytokine storms are established phenomena in AD; three main effector cells are implicated in this process: mast cells, central nervous system (CNS) microglia and astrocytes. CNS microglia are known to control neurogenesis. Hence, an overactivation of these cells would cause more newborn neurons to die by apoptosis, leading to a deficit in neurogenesis. All of these factors together or separately could be the cause of AD, which results at the end in memory loss and cognitive impairment.

**Figure 3 biomolecules-12-01409-f003:**
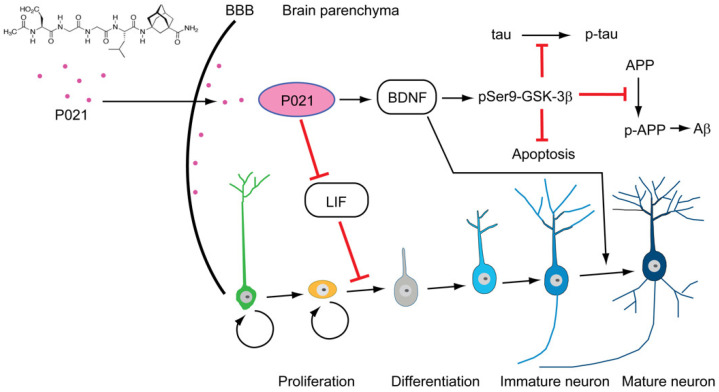
Mechanism of action of P021. P021 acts by competitively inhibiting leukemia inhibitory factor (LIF) signaling and increasing the expression level of BDNF. Together, the inhibition of LIF signaling and the increase in BDNF expression would inhibit stem cell proliferation and promote their development. BDNF has been shown to increase synaptic plasticity, neurogenesis, and cell survival. An increase in BDNF expression leads to an increase in the inhibitory phosphorylation of GSK3β at Serine 9. The prevention of Aβ and tau pathologies with P021 could be through the inhibition of GSK3β signaling. The inactivation of GSK3β would result in a reduction of the Aβ load and tau hyperphosphorylation. GSK3β is further known to promote apoptosis, and its inhibition would also lead to the inhibition of neurodegeneration.

**Table 1 biomolecules-12-01409-t001:** Neuropathological features of AD.

Neuropathological Features	Composition	References
**Amyloid plaques**	Composed of Aβ 40 and Aβ 42 fragments that result from the sequential cleavage of AβPP by the enzymes β-secretase and γ-secretase in neurons. There are two types of amyloid plaques: diffuse and dense core based on their morphology and their positive or negative staining with Thioflavin-S or Congo Red.	[28,29,30,31]
**Dense core plaques**	Dense core amyloid plaques stain with thioflavin-S and Congo Red and they are typically surrounded by dystrophic neuritis, reactive astrocytes and activated microglial cells, and are associated with synaptic loss.	[28,29,30,31]
**Diffuse plaques**	They are amorphous plaques with an undefined contour and amorphous amyloid deposits which are negatively stained with Congo red and thioflavin S.	[29]
**Amyloid β**	Amyloid β: a 40 or 42 amino acid peptide derived from amyloid precursor protein (APP) after its sequential cleavage by β- and γ-secretases.	[29]
**Cerebral amyloid angiopathy (CAA)**	It is the consequence of the deposition of Aβ in the vessel walls. The major constituent of CAA is the soluble form of Aβ (Aβ40)	[29]
**Neurofibrillary tangles (NFTs)**	They are primarily made up of paired helical filaments (PHFs) that are fibrils of 10 nm in diameter that form pairs with a helical tridimensional conformation at a regular periodicity of ~65 nm They are caused by the aggregation of the hyperphosphorylated tau in neurons of the misfolded tau that become extraneuronal (“ghost” tangles) when tangle-bearing neurons die.	[28,30]
**Neuropil threads**	They are axonal and dendritic segments formed by the aggregated and hyperphosphorylated tau that are usually associated with the NFT in AD.	[29]
**Granuovacuolar degeneration (GVD) and Hirano bodies**	GVD mainly formed by large double-membrane bodies with an unknown origin and significance. Usually detected in the cytoplasm of hippocampal pyramidal neurons of AD patients.	[31]
**Glial responses** **(Neuroinflammation)**	A significant positive correlation was reported between both astrocytosis and microgliosis and NFT burden but not between both reactive glial cell types and amyloid burden, which suggests that neuroinflammation is tightly linked to neurofibrillary degeneration.	[29,32]
**Neuronal and synaptic loss**	Neuronal loss is the major cause of cortical atrophy. Synaptic Loss contributes along with neuronal loss to cortical atrophy.	[11,29,33]
**α-synuclein positive Lewy bodies**	AD patients that present α-synuclein positive Lewy bodies exhibit acceleration in the disease process and a more aggressive and rapid cognitive decline compared to pure AD patients.	[34]
**Cognitive decline**	Episodic memory is the first area affected in the AD process, followed by impairment in the executive functions, apraxia, visuospatial navigation deficits, visuo-perceptive deficits, and semantic memory, which consequently results the full-blown dementia syndrome.	[29,35]

**Table 2 biomolecules-12-01409-t002:** Psychiatric symptoms of AD.

Psychiatric Symptoms	Prevalence	References
**Depression**	Its prevalence is around 20–50% in AD patients.	[151,152]
**Apathy**	Its prevalence could reach up to 80% Is the most common and persistent neuropsychological feature in AD	[152]
**Agitation, irritability and aggression**	Its prevalence is between 48% and 80% with symptoms that persistfor months and happen across all AD stages.	[153,154]
**Anxiety and phobia**	Prevalence was reported to range from 7.9% to 29.8%	[152,155]
**Psychotic symptoms** **(delusions and hallucinations)**	The prevalence is between 30–50% in AD. Hallucination was found to diagnose AD with 14% sensitivity and 99% specificity.	[152,156]
**Sleep disorders**	Common behavioral disturbances in AD. Prevalence between 25% to 50% of patients ~75% of patients sleep for extended periods during the day.	[152,157]
**Hypokinesia**	Could diagnose AD with 30% sensitivity and 99% specificity	[156]
**Paranoia**	Could diagnose AD with 15% sensitivity and 99% specificity	[156]
**Rigidity**	Could diagnose AD with 16% sensitivity and 100% specificity	[156]
**Tremors**	Could diagnose AD with 16% sensitivity and 96% specificity	[156]

**Table 3 biomolecules-12-01409-t003:** Current status of AD drugs, approved and under development.

Drug	Effects	References
Four cholinesterase inhibitors: **Donepezil** (Aricept™), **rivastigmine** (Exelon™), and **galantamine** (Razadyne™). **Tacrine**: No longer available on the market	Targets cholinergic innervations in the nucleus basalis.	[207]
One NMDA receptor antagonist: **Memantine** (Namenda™).	N-methyl-d-aspartate receptor antagonist (NMDA) that blocks glutamate from binding to its receptors. This prevents excessive excitotoxicity and neuronal cell death, which is thought to contribute to the pathogenesis of AD.	[207,208]
GV-971 (Oligomannate™), an oligosaccharide	Reduction of systemic inflammation and neuroinflammation - Approved in China	[207]
**Aducanumab**	First disease-modifying therapy (DMT). Became available on the market in 2021 for MCI due to AD and mild AD dementia An anti-amyloid monoclonal antibody Accelerated regulatory mechanism based on demonstration of amyloid plaque lowering	[207,209]
**Donanemab and lecanemab**	Monoclonal antibodies Under review by the US Food and Drug Administration (FDA).	[210,211]
Phase 3 clinical trials **31 agents**	21 DMTs (5 biologic and 16 small molecules)	[207]
Phase 2 clinical trials **82 agents**	71 DMTs (26 biologics and 45 small molecules).	[207]
Phase 1 clinical trials **30 agents**	27 DMTs (9 biologics and 18 small molecules)	[207]
**Antipsychotic drugs**		
**Acetylcholinesterase inhibitors**	May improve apathy, delusions and hallucinations, and less commonly improve aggression, depression, disinhibited behaviors, irritability or nocturnal disruption in patients with mild to moderate dementia	[212]
**Selective serotonin reuptake inhibitors (SSRIs)**	Effective in the management of depression and anxiety in people with dementia that cannot be treated by non-pharmacological interventions alone. For AD patients, citalopram was reported to decrease agitation and to likely improve other symptoms such as delusions, suggesting that it may have antipsychotic effect	[213,214]
**Antipsychotics** Risperidone, Quetiapine, Olanzapine	Have only a modest effect in managing the psychological symptoms that accompany AD and other neurodegenerative diseases The level of effectiveness of these drugs varies between patients.	[215]

## Data Availability

Not applicable.

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
