# Peer review of "Alzheimer’s Disease: Challenges and a Therapeutic Opportunity to Treat It with a Neurotrophic Compound"

_biomolecules, 2022, doi:10.3390/biom12101409_

Round 1

Reviewer 1 Report

I have some comments on the manuscript entitled “Alzheimer’s Disease: Challenges and a Therapeutic Opportunity to Treat it with a Neurotrophic Compound”.

1.       Write limitations of your study at the end of the abstract section.

2.       Elaborate your major objectives and approaches in the last paragraph of introduction section just before the heading “Ethiopathogenesis of AD”.

3.       Cognitive impairment also occurs in the case of Parkinson’s disease. How it is differed from AD. Some herbal plants like Mucuna pruriens and its bioactive components like ursolic acid and chlorogenic acid also improves cognitive impairment in PD. So first discussed Anti-Parkinsonian activity of Mucuna pruriens, ursolic acid and chlorogenic acid in MPTP intoxicated mouse model and relates its therapeutic efficacy in AD.

4.       Figures are very superficial. Some signalling related figures will be needed for your manuscript.

5.       One table regarding recent advancement in the therapeutics of AD will also be needed in your manuscript to enhance its quality.

6.       Complete editorial checking will be needed to correct the grammatical and punctuation mistakes.

Author Response

  1. Write limitations of your study at the end of the abstract section.

Response: Thank you! We have now added the limitations of our study at the end of abstract.

  1. Elaborate your major objectives and approaches in the last paragraph of introduction section just before the heading “Ethiopathogenesis of AD”.

Response: The point is well taken. We have now added a paragraph concerning the major objectives and approaches of the present work in the last section of the introduction before the heading “Ethiopathogenesis of AD”.

  1. 3.       Cognitive impairment also occurs in the case of Parkinson’s disease. How it is differed from AD. Some herbal plants like Mucuna pruriens and its bioactive components like ursolic acid and chlorogenic acid also improves cognitive impairment in PD. So first discussed Anti-Parkinsonian activity of Mucuna pruriens, ursolic acid and chlorogenic acid in MPTP intoxicated mouse model and relates its therapeutic efficacy in AD.

Response: Thank you! We appreciate the reviewer’s comment but the required addition is out of the scope of our review article and hence is not included.

  1. Figures are very superficial. Some signaling related figures will be needed for your manuscript.

Response:The point is well taken. However, Figure.3 represents the signaling mechanism of P021.

  1. One table regarding recent advancement in the therapeutics of AD will also be needed in your manuscript to enhance its quality.

Response: The point is well taken. We have now added a table concerning the recent advancements in the therapeutics of AD (Table.3).

  1. Complete editorial checking will be needed to correct the grammatical and punctuation mistakes

Response: Thank you! We have rechecked  paper for any typographical and grammar mistakes.

Reviewer 2 Report

The Authors have developed a peptidergic compound named P021 which is composed of four amino acids derived from a biologically active region of ciliary neurotrophic factor (CNTF) and is C-terminally adamantylated to increase its lipophililicityand decrease its degradation by exopeptidases.

In the present review, the Authors evaluated the recent studies that were conducted in Their laboratories to evaluate its potential as a drug for the treatment of Alzheimer’s disease (AD) and other neurodegenerative diseases.

Overall, I found this paper very interesting, timely, well written and scientifically sound. I have only some minor suggestions aimed to improve the high quality of the review and these are otlined below:

1) Please, add the chemical structure of such Neurotrophic Compound P021.

2) Even if AD is a well known disease, I suggest the Authors to add a Table with common neuropathological findings and how they relate to AD' symptoms, including BPSD.

3) Besides, I suggest to add a brief note on the current state-of-art in the treatment od AD (and also in the Table above suggested, if possible).

Author Response

  1. Please, add the chemical structure of such Neurotrophic Compound P021.

Response:Thank you! The chemical structure of P021 is shown in Figure.3.

  1. Even if AD is a well known disease, I suggest the Authors to add a Table with common neuropathological findings and how they relate to AD' symptoms, including (BPSD).

Response: We have now added a table concerning common neuropathological and psychological features of the disease (Tables.1 and 2).

  1. Besides, I suggest to add a brief note on the current state-of-art in the treatment of AD (and also in the Table above suggested, if possible).

Response: We have now added a table concerning the latest data about the approved drugs for AD and the drugs that are under phase 1, phase 2 and phase 3 clinical trials (Table.3). We have also added a paragraph about the current state-of-art in the treatment of AD, before the Conclusion section.